# Topological order and entanglement dynamics in the measurement-only XZZX quantum code

Kai Klocke[1] and Michael Buchhold[2]

[1]*Department of Physics, University of California, Berkeley, California 94720, USA*
[2]*Institut für Theoretische Physik, Universität zu Köln, D-50937 Cologne, Germany*
(Dated: today)

We examine the dynamics of a $(1+1)$-dimensional measurement-only circuit defined by the stabilizers of the [[5,1,3]] quantum error correcting code interrupted by single-qubit Pauli measurements. The code corrects arbitrary single-qubit errors and it stabilizes an area law entangled state with a $D_2 = \mathbb{Z}_2 \times \mathbb{Z}_2$ symmetry protected topological (SPT) order, as well as a symmetry breaking (SB) order from a two-fold bulk degeneracy. The Pauli measurements break the topological order and induce a phase transition into a trivial area law phase. Allowing more than one type of Pauli measurement increases the measurement-induced frustration, and the SPT and SB order can be broken either simultaneously or separately at nonzero measurement rate. This yields a rich phase diagram and unanticipated critical behavior at the phase transitions. Although the correlation length exponent $\nu = \frac{4}{3}$ and the dynamical critical exponent $z = 1$ are consistent with bond percolation, the prefactor of the logarithmic entanglement growth may take non-integer multiples of the percolation value. Remarkably, we identify a robust transient scaling regime for the purification dynamics of $L$ qubits. It reveals a modified dynamical critical exponent $z^* \neq z$, which is observable up to times $t \sim L^{z^*}$ and is reminiscent of the relaxation of critical systems into a prethermal state.

## I. INTRODUCTION

The competition between non-commuting operators lies at the heart of quantum mechanics, e.g., inducing correlations and frustration in quantum many-body systems, and forming the cornerstone of quantum technology, including quantum computation and quantum error correcting codes (QECC). The latter has recently been scrutinized from the viewpoint of monitored quantum circuits and measurement-induced entanglement transitions. In a quantum circuit, frequent local measurements, which do not commute with the generators of the unitary dynamics induce a phase transition in the dynamics of entanglement.[1–3] This phenomenon has been observed in random quantum circuits, where the unitary evolution is generated by Clifford[2–17] or Haar[1,3,18–26] random gates and in Hamiltonian systems, where the unitary time evolution is continuous.[27–39] Due to the inherent randomness of the measurement process, the entanglement phase transitions do not manifest on the level of ensemble averaged, local order parameters but only in higher moments, or replicas of the state,[11,20,31,40,41] a feature shared in common with many topological phase transitions.

Comparable entanglement transitions happen in measurement-only quantum circuits, where the evolution of the wave function is exclusively generated by projective measurements. Frustration is induced when the measured operators are drawn from distinct sets of local, incommensurate operators. This may lead to, e.g., the build up of a volume law entangled state due to measurements and induce an entanglement transition into an area law entangled state[42,43] or a phase transition between two area law entangled states with different topological order.[44,45] Measurement-only dynamics are naturally related to the idea of quantum error correcting codes, where we may imagine a competition between the mea-

surement of parity check operators for a particular error correcting code, and single-qubit measurements mimicking the adverse influence of the environment.[41,44]

Although a realistic error correcting scenario involves intermediate gates corresponding to both the desired computation and the corrections made based on the parity check syndrome, measurement-only circuits offer a minimal model for understanding entanglement dynamics in this setting. The measurement-only version of the quantum repetition code, for instance, displays an entanglement phase transition corresponding to two-dimensional bond percolation,[4,41,44] which signals the spoiling of the logical qubit due to single-Pauli errors. The repetition code represents the most elementary QECC, correcting exclusively "classical" bit flip errors, while more advanced codes are required in order to correct arbitrary single-qubit errors. For the latter, one may expect a more nuanced, genuine quantum dynamics due to the enhanced number of non-commuting measurements. In $(2 + 1)$-dimensions, for instance, this was confirmed recently in the measurement-only variant of the toric code, where a volume law phase is observed once arbitrary single-qubit errors are allowed.[42] Due to the recent progress in implementing stabilizer codes in near-term quantum devices,[46,47] probing entanglement transitions in experimentally relevant stabilizer codes and understanding their relation to the capability of performing fault tolerant error correction in quantum circuits appear as promising near term goals to advance QECCs.

In this work we examine the measurement-only variant of the $(1 + 1)$-dimensional [[5,1,3]] QECC, which is capable of correcting arbitrary single-qubit errors. In our setting, the "errors" are represented by single Pauli operator measurements (either $X, Y, Z$). The code space hosts a $D_2$ symmetry protected topological (SPT) order and a $\mathbb{Z}_2$ bulk symmetry breaking order, which can both

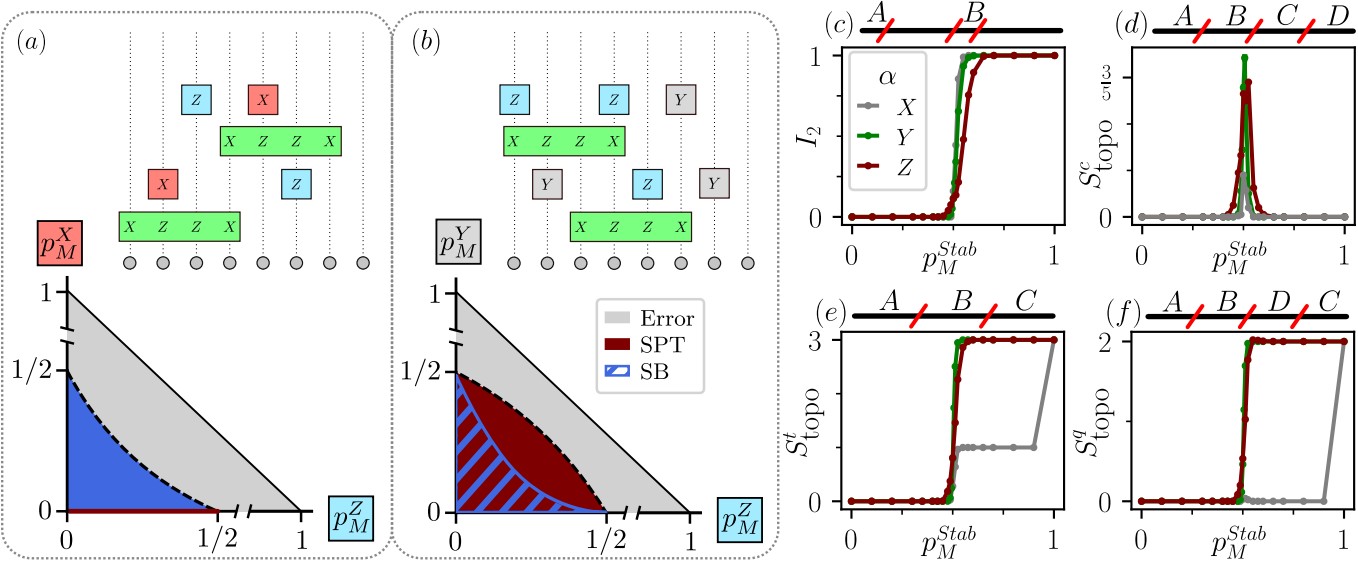

Figure 1. (a) Phase-diagram for a measurement-only evolution driven by measurement of stabilizers $XZZX$ and single-site Pauli operators $X$ and $Z$, with probabilities $p_M^{Stab}$, $p_M^X$, and $p_M^Z$, respectively. For weak error measurement, the system falls into an area-law phase with two-fold degenerate bulk-symmetry (blue). With only $Z$ measurements, the stabilizer phase preserves an SPT order (maroon). (b) Phase-diagram now allowing only $Y$ and $Z$ error measurements. Here the SPT order is preserved throughout the entire stabilizer phase, whereas the bulk-symmetry is broken at weaker error probability $p_M^Z$, $p_M^Y$. (c)-(f) Different entanglement measures calculated for the depicted subsystems reveal the transition from the error phase to the stabilizer phase when only one flavor of Pauli measurement is allowed. (c) The mutual information $I_2(A,B)$ between well-separated regions $A$ and $B$ probes the existence of globally shared qubits. (d) The conditional mutual information $S_{\text{topo}}^c$ reveals the critical point and log-law coefficient $\tilde{c}$. (e) The generalized topological entanglement entropy $S_{\text{topo}}^t$ probes both SPT order and symmetry breaking. (f) Similarly, $S_{\text{topo}}^q$ probes only the SPT order.

be broken by the Pauli errors at a sufficiently large measurement rate. The case of unique single-qubit errors (e.g., only Pauli-$Z$ or only Pauli-$X$) turns out to be, up to minor modification, equivalent to the quantum repetition code, and displays an entanglement phase transition in the universality class of two-dimensional (2D) bond percolation. The scenario changes when multiple single-qubit errors are allowed (e.g., Pauli-$X$ and Pauli-$Z$).

In the presence of multiple, incommensurate errors, the bulk symmetry and SPT order can be broken at separate, nonzero single-qubit measurement rates, with the order at which the two transitions occur being sensitive to the allowed type of errors. In both cases, however, only one of the transitions appears to display critical behavior, i.e., shows a logarithmic growth of the entanglement entropy. Depending on the types of measurements, the critical point shifts to larger or smaller error rates and we find that the logarithmic entanglement entropy scaling can be enhanced along the transition by measurement frustration between single-qubit errors. A strong effect of incommensurate errors is observed in the dynamics of entanglement. The entanglement growth in a pure state and the entanglement fluctuations in the asymptotic state both confirm a dynamical critical exponent $z = 1$. The evolution of an initial mixed state, however, reveals the emergence of a second, transient scaling regime in the purification dynamics. It exhibits a distinct dynamical exponent $z^* \neq z$ that is sensitive to the

allowed error measurements and persists up to time scales $t \sim L^{z^*}$ ($L$ is the system size), when the number of unpurified qubits is $\mathcal{O}(1)$. Overall, the additional measurement frustration caused by incommensurate single-qubit errors gives rise to a diverse phenomenology, including previously unanticipated dynamical scaling and topological phase transitions.

The paper is organized as follows. In Sec. II we provide a brief review of the essential features of the [[5,1,3]] code, including the associated SPT order and symmetry breaking order in the code space. We then establish the entanglement measures in Sec. III, which we will use to characterize the entanglement transition and the topological order in each phase. To set the stage, we focus on a single type of Pauli error in Sec. IV and show that this scenario, up to minor modifications, is reminiscent of the repetition code. Finally in Sec. V, VI, we examine the more diverse phenomena which arise from multiple competing error measurements.

## II. THE XZZX CIRCUIT MODEL

We examine the entanglement dynamics in a measurement-only variation of the [[5,1,3]] QECC. This is the smallest QECC capable of correcting an arbitrary single qubit error, and in this sense it is the smallest true 'quantum' code. In the [[5,1,3]] code, a single logical

qubit is encoded across 5 physical qubits, with stabilizers defined by the Pauli strings $M_i = X_i Z_{i+1} Z_{i+2} X_{i+3}$, $i = 1, ..., 4$ and periodic boundary conditions ($\prod_{i=1}^{4} M_i = M_5$). The code space is the two-fold degenerate subspace, for which all stabilizers $M_i = +1$. The logical operators for the encoded qubit in this subspace are $\bar{X} = XXXXX$ and $\bar{Z} = ZZZZZ$.

We extend this code to an arbitrary number ($L$) of qubits and take *open boundary conditions* (OBC), for which $i = 1, \ldots, L - 3$. From hereon we refer to this extended version of the [[5,1,3]] code as the XZZX-code. Closely related XZZX models in one and two spatial dimensions have been considered in the error correcting context,[48,49] where they exhibit a robust error threshold for single-qubit Pauli noise and can be modified for maximal code distance with biased noise channels. For OBC we can define three additional pairs of global operators $(\bar{X}_l, \bar{Z}_l)$ with $l = 1, 2, 3$ and $\bar{Z}_l = \prod_{i=0}^{L/3} Z_{3i+l}$ and equivalent for $\bar{X}_l$. These $\bar{Z}_l$ mutually commute with each other and with all the stabilizers $M_i$[50]. This gives rise to an 8-fold degenerate code space, which hosts both the logical qubit (equivalent to a $\mathbb{Z}_2$ bulk symmetry breaking order) and also a $D_2 = \mathbb{Z}_2 \times \mathbb{Z}_2$ symmetry protected topological (SPT) order.[51,52] In the language of fermions, the $\bar{Z}_l$ operators correspond to the sublattice fermion parity.

The origin of the $D_2$ SPT order and the logical qubit can be understood by observing that each stabilizer may be written as the product of smaller overlapping stabilizers, $M_i = (X_i Y_{i+1} X_{i+2})(X_{i+1} Y_{i+2} X_{i+3})$, where all $X_i Y_{i+1} X_{i+2}$ commute with all $M_i$.[52] Then fixing a measurement outcome for each stabilizer $M_i$, the logical qubit arises from the two-fold degeneracy in assigning expectation values to all $X_i Y_{i+1} X_{i+2}$. Similarly, the product of all stabilizers $M_i$ is a Pauli string $X_1 Y_2 X_3 \cdots X_{L-2} Y_{L-1} X_L$. The isolated $XYX$ strings at each end anticommute with the total parity operator $P = \bar{Z}_1 \bar{Z}_2 \bar{Z}_3$, thereby generating the $D_2$ symmetry. The $\mathbb{Z}_2$ bulk symmetry associated to the logical qubit and the $D_2$ SPT order can be separately probed and broken by the measurement of appropriate Pauli operators, which we explore in the remainder of this work.

In our measurement-only circuit, the dynamics is generated by projective measurement of (i) the stabilizers $M_i$ of the XZZX code and (ii) single site Pauli operators $X_i, Y_i, Z_i$. We refer to the single site measurements as "errors", which aim to break the globally encoded qubit. The circuit evolution consists of alternating layers.[44] On even layers, the stabilizers $M_i$ are measured, each with probability $p_M^{Stab}$. On odd layers, single site errors are applied with probability $1 - p_M^{Stab}$. If an error happens at site $i$ then the corresponding Pauli operator $\alpha_i = X_i, Y_i, Z_i$ is chosen with probability $q_\alpha \in [0, 1]$ such that $q_X + q_Y + q_Z = 1$. Our unit of time will be the number of layers and a steady state is typically reached after $\mathcal{O}(L)$ steps. We apply different entanglement measures to map out a family of measurement-induced phase transitions, as a function of the error probabilities. Since all measurements correspond to the Pauli group, the circuit

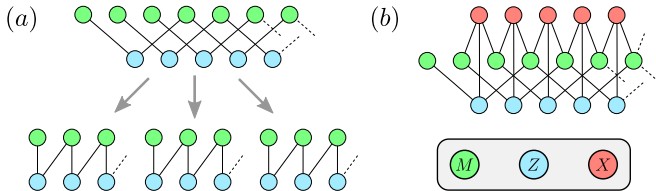

Figure 2. Measurement frustration graph. (a) With only stabilizer and $Z$ measurements, the graph consists of three disconnected bipartite subgraphs. This yields a tripling of $\tilde{c}$ compared to the case with only $X$ errors, which corresponds to a single one of the bipartite subgraphs. (b) With both $Z$ and $X$ error measurements allowed, the graph is no longer bipartite nor does it contain any disconnected subgraphs. The critical measurement strength is thus shifted away from $p_M^{Stab} = \frac{1}{2}$,

can be efficiently simulated by tracking the set of generators $\mathcal{G} = \langle S_1, \ldots, S_L \rangle$ of the stabilizer group $\mathcal{S}$ for the state.[53,54]

The circuit dynamics exhibits a competition between stabilizer measurement and errors. When either type of measurements dominates, the steady state is stabilized by a set of (quasi-) local Pauli operators and therefore obeys an entanglement area law. The correlations and entanglement properties are then determined by the class of operators, i.e. stabilizers $M_i$ or errors $\{X_i, Y_i, Z_i\}$, which dominantly stabilize the steady state. We denote the phase, which is dominated by stabilizer measurements *stabilizer phase* and the one dominated by Pauli measurements as *error phase*. A continuous phase transition separates the stabilizer phase from the error phase, roughly at values where the measurement rates are comparable ($p_M^{Stab} \simeq 0.5$). At the transition one finds a logarithmic entanglement growth and several additional features, depending on the errors, which we characterize in this paper.

## III. MEASURES OF ENTANGLEMENT

In order to characterize the dynamics and the steady state in the circuit, we use a combination of different entanglement measures, which we briefly overview below. All measures are based on the von Neumann entanglement entropy $S_A = -\text{Tr}(\rho_A \log_2 \rho_A)$, where $\rho_A$ is the reduced density matrix for a (sub-)region $A$.[55] In the stabilizer formalism, $S_A$ is determined by the number of well-defined, independent stabilizers acting only on $A$.[56,57]

In order to distinguish between the two area-law phases, we consider the *mutual information* $I_2(A, B)$ between separated regions $A$ and $B$ defined as

$$I_2(A, B) = S_A + S_B - S_{AB} \tag{1}$$

where $AB \equiv A \cup B$. Throughout this work we take $A = \{1, 2, \ldots, \frac{L}{8}\}$ and $B = \{\frac{L}{2}, \frac{L}{2} + 1, \ldots, \frac{5L}{8}\}$, as shown in Fig. 1c. $I_2(A, B)$ corresponds to the total number of independent stabilizers on $A$ and $B$ which are *not* in-

dependent on $AB$. When stabilizer measurements dominate, they generate extensive clusters, leading to single nonlocally encoded logical qubit and potentially edge modes. The bulk-encoded qubit yields a mutual information of exactly $I_2(A, B) = 1$. In a corresponding error correction protocol, this is equivalent to preserving the information of an initial state $|\psi\rangle = \alpha\,|\bar{0}\rangle + \beta\,|\bar{1}\rangle$, though scrambled, in the wave function.

The 8-fold ground state degeneracy of the XZZX-code with OBC leads to richer physics, including the discussed topological order, than can be probed through the mutual $I_2(A, B)$. To this end, we consider the more general conditional mutual information,

$$I_2(A, C|B) = I_2(A, BC) - I_2(A, B), \qquad (2)$$

for a partitioning of the system into at least three regions $A$, $B$, $C$. Depending on the partitioning, $I_2(A, C|B)$ acts as a generalized topological entanglement entropy, which distinguishes different types of topological order.[45,51,58] We use three distinct ways to partition the system, each of which is depicted in Fig. 1(d-f). The corresponding conditional mutual information and their interpretations are given below:

- $\mathbf{S_{topo}^t}$ - For a partitioning of the system into three contiguous regions $A$, $B$, $C$ (see Fig. 1e), the conditional mutual information $I_2(A, C|B)$ probes the non-local information shared between well-separated regions.[51] In particular it counts all the nontrivial global operators stabilizing the state. Here this corresponds to the two possible generators of the $D_2$ SPT order and the single generator of the $\mathbb{Z}_2$ bulk symmetry, yielding a maximum value of $\mathbf{S_{topo}^t} = 3$.

- $\mathbf{S_{topo}^q}$ - The conditional mutual information evaluated on a partitioning of the system into four equally sized regions $A$, $B$, $D$, $C$ such that $C$ is spatially separated from $A$ and $B$ (see Fig. 1f). Since $I_2(A, C|B)$ involves only the reduced density matrix on $ABC$, bulk subsystem $D$ is traced out, leaving $S_{topo}^q$ insensitive to bulk symmetries. Instead $S_{topo}^q$ counts only the symmetry generators carried at the boundaries (i.e. from SPT order)[27,45] and so takes a maximal value of 2.

- $\mathbf{S_{topo}^c}$ - For a partitioning of the system into four contiguous and equally sized regions $A,B,C,D$ (see Fig. 1d), all boundary and volume terms cancel, leaving only a possible contribution from a log-law term. For a log-law with $S_A = \frac{\tilde{c}}{3}\log_2\left(\frac{L}{\pi}\sin\left(\frac{\pi|A|}{L}\right)\right)$, $S_{topo}^c = \frac{\tilde{c}}{3}$. This provides a convenient means by which to identify the critical point and extract the entanglement scaling $\tilde{c}$ from a single quantity.

In addition to the stationary entanglement in the steady state, we examine the *dynamics* of entanglement.

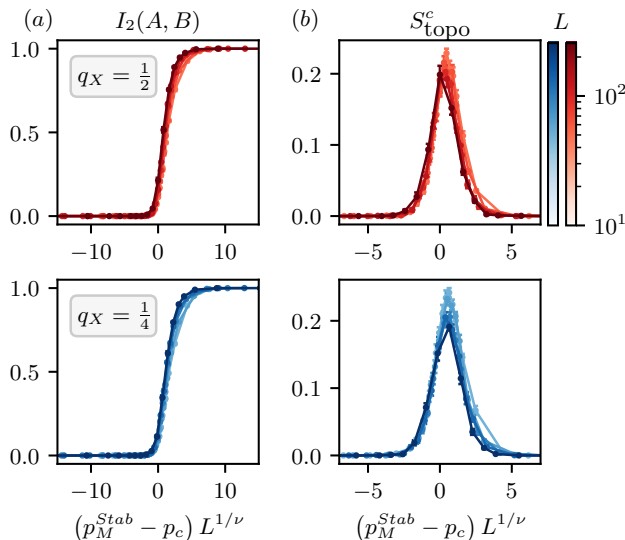

Figure 3. $X$ and $Z$ errors: Data collapse for (a) the mutual information $I_2(A, B)$ between well separated segments $A$ and $B$, and (b) $S_{topo}^c$ for $L = \{48, 64, 96, 128, 192, 256\}$. The top row shows data for $q_X = q_Z = \frac{1}{2}$, where $p_c \approx 0.555$ and $\nu \approx \frac{4}{3}$. Similarly the bottom row shows data for $q_X = \frac{1}{4}$, $q_Z = \frac{3}{4}$, where $p_c \approx 0.562$ and $\nu \approx \frac{4}{3}$. While the critical point shifts as a function of $q_\alpha$, the exponent $\nu$ remains consistent with the percolation value.

We determine the following dynamical measures: (i) the growth of the half-chain entanglement entropy $S_{L/2}(t)$ starting from an initial product state, (ii) the power-spectrum $|S_{L/2}(f)|$ of the fluctuations of $S_{L/2}(t)$ in the steady-state, and (iii) for a mixed state, we consider the *residual entropy* $\langle S_L\rangle = -\operatorname{Tr}(\rho\log_2\rho)$, which counts the number of mixed qubits left to be purified. For a single trajectory, the residual entropy is the logarithm of the purity, and provides a useful metric for the dynamics of purification in the circuit.

## IV.  UNIQUE PAULI ERRORS

When only one type of single-site Pauli error is allowed (i.e. some $q_\alpha = 1$), the dynamics in the code is, up to minor modifications, equivalent to the previously studied quantum repetition code.[41,44] This can be understood from the so-called *measurement frustration graph*.[43] For a given realization of errors and stabilizers, the graph consists of vertices for every operator which can be measured, and edges connecting any anticommuting operators. Associating every vertex with a weight set by the measurement probability, we may infer properties of the circuit and transition from this graph. For a single type of onsite Pauli (either X, Y, Z), the graph is bipartite (e.g., for $Z$ see Fig. 2a), and thus invariant under exchange of the two subgraphs when $p_M^{Stab} = \frac{1}{2}$. This pins the critical point precisely at $p_M^{Stab} = \frac{1}{2}$ for any unique

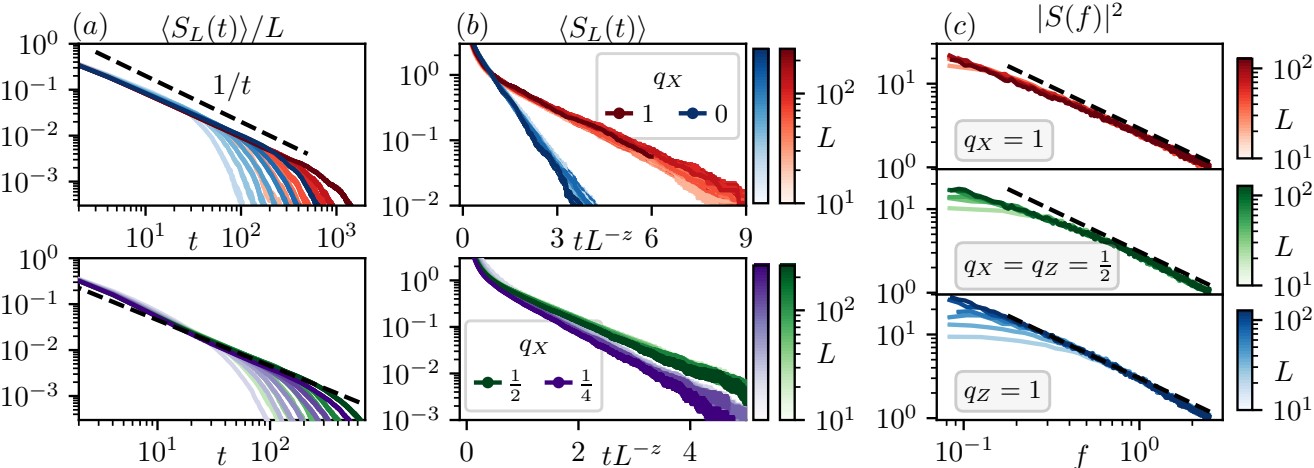

Figure 4. Asymptotic entanglement dynamics revealing an exponent $z = 1$ at criticality with $q_X + q_Z = 1$. (a) Power-law decay of the residual entropy density $\langle S_L(t)\rangle/L$ at intermediate times for $q_X = 0$ (blue), 1 (red), $\frac{1}{2}$ (green), and $\frac{1}{4}$ (purple). Decay is consistent with $1/t$ (dashed black line), corresponding to $z = 1$. (b) Data-collapse with $z = 1$ of the residual entropy at late times, where it decays exponentially fast. For $q_X = 0, 1$, the relative decay rates corresponds to the relative magnitudes of $\tilde{c}$. (c) Power-spectrum $|S(f)|^2$ for fluctuations in the half-chain entanglement entropy in the steady state of the pure-state dynamics. Observation of $1/f$ noise is consistent with $z = 1$. The frequency feature associated with the alternating circuit structure has been truncated here. Comparable results are obtained with a circuit in which all measurements may occur in each layer (as in Ref.[45]).

type of errors. Collapsing the entanglement measures to a single function $F\left(\left(p_M^{Stab} - p_c\right)L^{1/\nu}\right)$ yields a critical exponent consistent with $\nu = \frac{4}{3}$, expected from 2D bond percolation[59]. See App. A for details on the scaling collapse. For $\alpha = X, Z$ the mapping from the circuit dynamics to a 2D bond-percolation problem is equivalent to the known mapping for the repetition code[44] in terms of the colored cluster model.[41] Depending on the type of error some characteristics vary at the transition, which we discuss below.

*Only X errors, $q_X = 1$.* – Here the dynamics has a one-to-one correspondence to the repetition code.[44] Due to the commutation relations, the stabilizers are effectively reduced to $M_i \equiv Z_{i+1}Z_{i+2}$ for all $i$. With open boundary conditions, the first and last site are never acted upon by the stabilizers. Thus any SPT order will be immediately destroyed by measuring $X_1$ or $X_L$, which anticommute with the global operators $\bar{Z}_l$ for OBC. This is reflected in Fig. 1e,f where both $S_{\text{topo}}^q$ and $S_{\text{topo}}^t$ are reduced by 2 for any non-zero $p_M^X$. The two-fold degeneracy associated to the bulk-symmetry remains, however, intact at non-zero $p_M^X$ and we find $S_{\text{topo}}^t = 1$ throughout the stabilizer phase (see Fig. 1f). The critical point separating the two area-law phases features a logarithmic entanglement scaling, reflected by the narrow peak in $S_{\text{topo}}^c$. In particular, $\tilde{c} = \frac{3\sqrt{3}\log(2)}{2\pi}$, as in 2D percolation and related measurement-only circuits.[41,44,45]

The relation to 2D bond percolation can be summarized as follows. Measuring a stabilizer $M_i$ puts sites $i+1$ and $i+2$ into the same quasi-GHZ state. Subsequent sta-

bilizer measurements thus nucleate, grow, or merge clusters, while measuring $X_i$ will remove site $i$ from any cluster. The spread of entanglement corresponds to the development of extensive clusters which may connect (and thus entangle) distant regions of the system. Two qubits are in the same cluster state if and only if there exists a connected path of stabilizer measurements unbroken by errors in the circuit's history.

*Only Z errors, $q_Z = 1$.* – Here, the stabilizers reduce to $M_i \equiv X_iX_{i+3}$ on all reachable states. The measurement frustration graph then consists of three disconnected components, each of which realizes a copy of the repetition code (see Fig. 2a). Each of the three copies yields a percolation transition on the corresponding sublattice, with measurement of a stabilizer $M_i$ now placing sites $i$ and $i + 3$ into the same quasi-GHZ cluster. The entanglement entropy is the sum of the three copies, giving three times the percolation value $\tilde{c} = 3\frac{3\sqrt{3}\log(2)}{2\pi}$ (see Fig. 1d). Since $Z$ measurements commute with the global operators $\bar{Z}_l$, the SPT order remains unbroken by nonzero $p_M^Z$ and only vanishes at the transition. As such, $S_{\text{topo}}^q = 2$ and $S_{\text{topo}}^t = 3$ throughout the stabilizer phase. Similarly, for the $XZX$ cluster model with $Z$-only errors, two copies of the repetition code form a stabilizer phase with SPT order.[45]

*Only Y errors, $q_Y = 1$.* – Concerning the topological properties, this setup closely resembles that found with $Z$ measurements. Since $Y_1$ anticommutes with both $M_1$ and $\bar{Z}_1$, measuring $Y_1$ will not remove the global operator and SPT order survives throughout the entire stabilizer phase. Moreover, if one considers a particular (bulk)

symmetry sector, where the effective stabilizers take the form $XYX$, we see that $Y$ measurements will not lift the SPT order within this sector. The measurement frustration graph is bipartite but does not easily factorize and therefore a direct mapping to the repetition code is not available. As a result, the quasi-GHZ cluster picture is necessarily distinct from the repetition code. For an initial state $\mathcal{G} = \langle Y_1, \ldots, Y_L \rangle$, measuring the stabilizer $M_1$ yields $\mathcal{G} = \langle M_1, Y_1 Y_2, Y_1 Y_3, Y_1 Y_4, Y_5, \ldots, Y_L \rangle$, forming the overlapping clusters stabilized by $Y_1 Y_2, Y_1 Y_3, Y_1 Y_4$. More generally, measuring a stabilizer $M_i$ will put site $i$ into three clusters, with sites $i+1$, $i+2$, and $i+3$, while measuring $Y_i$ removes site $i$ from all clusters. As clusters grow, this picture becomes more complicated, and measuring a stabilizer $M_i$ will merge only clusters which anticommute with $M_i$. The failure of the frustration graph to factorize into copies of the repetition code corresponds to the formation of *overlapping* cluster states, altering the entanglement structure. Nonetheless, at the transition, $S^c_{\text{topo}}$ reveals a log-law coefficient consistent with four copies of percolation, $\tilde{c} = 4 \frac{3\sqrt{3}\log(2)}{2\pi}$. This is in line with the observation that (overlapping) cluster states remain a good description of the entanglement structure and that every stabilizer $M_i$ anticommutes with four different $Y_j$.

## V. $X$ AND $Z$ ERRORS

In the case of $X$ and $Z$ errors ($q_X + q_Z = 1$), the measurement frustration graph, Fig. 2b, is no longer bipartite. This leads to observable consequences both for the static critical behavior as well as for the dynamics. The critical point of the phase transition is shifted away from the value $p_c = 0.5$, which was generically observed for unique measurements, to larger values, with a maximum value of $p_c \approx 0.56$ when $q_X = \frac{1}{4}$. Along the critical line we find that $\nu \approx \frac{4}{3}$ (see Fig. 3), consistent with a percolation transition. The most drastic consequence is, however, observed in the dynamics, where a transient but robust dynamical critical exponent $z^* < 1$ is found.

The steady state phase diagram for $X$ and $Z$ errors is shown in Fig. 1a. Similar to the case of $q_X = 1$, for any nonzero probability of $X$ errors, the SPT order remains broken in the stabilizer phase, yielding $S^q_{\text{topo}} = 0$ and $S^t_{\text{topo}} = 1$. Furthermore, for $q_Z < 1$, due to the immediate coupling of the three sublattices by $X$ errors, the log-law coefficient $\tilde{c}$ drops to the value $\tilde{c} = \frac{3\sqrt{3}\log(2)}{2\pi}$, which was found for a single copy of the percolation transition. This value jumps discontinuously at $q_Z = 1$ (see App. C).

We note that the entanglement transition is equally well reproduced by examining the steady-state residual entropy $\langle S_L(t \to \infty) \rangle$ resulting from purification of a maximally mixed initial state (see App. C). In the error phase $\langle S_L \rangle$ vanishes since Pauli measurements on each site fully purify the state. On the other hand, $\langle S_L \rangle$ remains nonzero in the stabilizer phase, counting the num-

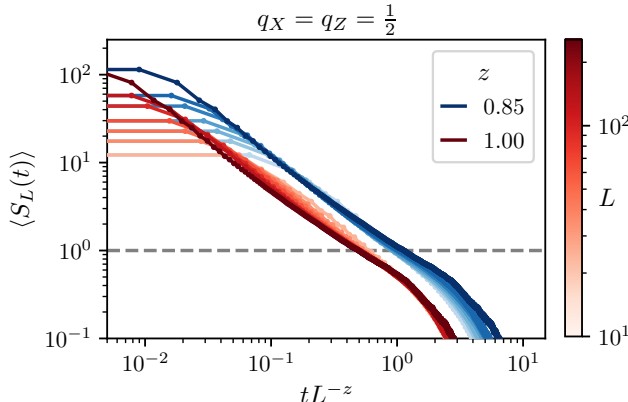

Figure 5. Residual entropy versus rescaled time $tL^{-z}$ for $q_X = q_Z = \frac{1}{2}$ at the critical point $p_c = 0.56$ with system sizes $L \in \{24, 36, 48, 64, 96, 128, 256\}$. We contrast the fitted exponent $z^* = 0.85$ (blue) with the asymptotic $z = 1$ (red), highlighting that the dynamics in the transient regime indeed exhibit an anomalous exponent $z^* \neq 1$. This regime terminates when the number of qubits remaining to be purified is of order 1 (dashed gray lines at 1). Moreover, we find that this scaling persists in the neighborhood of the critical point and that the exponent $z^* \neq 1$ is robust against variations in $p_M^{Stab}$.

ber of intact global operators which remain mixed, and so is similar to $S^t_{\text{topo}}$.

Next, we examine the dynamics at the critical point of the entanglement transition with $X$ and $Z$ errors and we focus on the dynamical critical exponent $z$. If the critical point corresponds to percolation, one expects $z = 1$. This value is confirmed in three different dynamical regimes: (i) the entanglement growth starting from an initial pure state, (ii) the asymptotic entanglement fluctuations in a pure state and (iii) the asymptotic purification dynamics starting from a maximally mixed state. However, we also detect a new, transient scaling regime where the dynamics reveals a robust dynamical critical exponent $z^* < 1$. This scaling regime is absent if only $X$ or $Z$ errors are present (i.e., $q_X = 1$ or $q_Z = 1$) and describes the purification of an initial mixed state up to the number of unpurified qubits is $\mathcal{O}(1)$. It thus dominates an extensive time regime in the thermodynamic limit $L \to \infty$.

*Entanglement growth in a pure state.* – Starting from an initial product state, the half-chain entanglement entropy grows logarithmically in time like $S_{L/2} \sim \frac{\tilde{c}_t}{3} \log_2(t)$. The dynamical exponent $z$ can be found by comparing the rate of entanglement growth in time and space so that $z = \tilde{c}/\tilde{c}_t$. In either limit $q_X, q_Z = 1$, one finds $z = 1$ owing to the exact mapping between the circuit evolution and classical 2D bond percolation. This is confirmed in App. C by the numerical simulations, and moreover we find that for all $q_X + q_Z = 1$ the entanglement growth from an initial pure state is consistent with $z = 1$.

*Asymptotic entanglement fluctuations in a pure state.* – Starting again from a pure state, we evaluate the

temporal fluctuations of $S_{L/2}(t)$ in the steady state. These persistent fluctuations are caused by measurements which break and restore stabilizers crossing between the left and right half of the system, and can be used to evaluate the dynamical critical exponent.[60] We numerically compute the power-spectrum $|S(f)|^2$ of the entanglement fluctuations and find that it exhibits a characteristic $1/f$ pattern for all $q_X + q_Z = 1$ at the critical point (see Fig. 4c), again yielding $z = 1$.[60]

*Asymptotic purification dynamics from a mixed state.* – The dynamical critical exponent can also be determined by studying the purification of an initial maximally mixed state $\rho \sim \mathbb{1}$ due to measurements.[5] We compute the time evolution of the residual entropy $\langle S_L(t) \rangle$. Its asymptotic evolution at late times is expected to follow an exponential decay $\sim e^{-\gamma_L t}$, with a rate that scales with the system size as $\gamma_L \sim L^z$. For $q_X = 1$ and $q_Z = 1$ the late time residual entropy scales exactly as $\langle S_L(t) \rangle \sim \exp\left(-\tilde{c}t/L\right)$.[43,61] At large times, where the average number of unpurified qubits is $\ll 1$, we observe a collapse of the purification data after rescaling $t \to tL^{-1}$, see Fig. 4b, for system sizes up to 256. This further confirms $z = 1$ in the asymptotic state.

*Transient dynamical critical scaling exponent $z^* < 1$.* – In a transient time regime, the residual entropy density $\langle S_L(t) \rangle/L$ is independent of system size and decays as a power-law. This gives another direct means to extract the dynamical critical exponent. For $q_X = 1$ and $q_Z = 1$ this unambiguously yields $z = 1$. However, when both $X$ and $Z$ errors are present simultaneously we find a transient time regime, which exhibits different scaling behavior. In this case, a scaling collapse of the purification data is obtained by rescaling time with a different critical dynamical exponent $t \to tL^{-z^*}$. This scaling collapse is robust, present up to times $t \sim L^{z^*}$, and works for all system sizes and for values of $p_M^{\text{Stab}}$ in the neighborhood of the critical point (see Fig. 15). For $q_X, q_Z > 0$ the dynamical critical exponent in the transient regime turns out to be smaller than the steady state value $z^* < z$, e.g. for $q_X = q_Z = \frac{1}{2}$ it is $z^* \approx 0.85$ as shown in Fig. 5. While numerical simulations indicate $z^*$ is insensitive to the value of $p_M^{\text{Stab}}$, it depends explicitly on $q_X, q_Z$. We find that it does *not* take on a universal value but rather varies continuously through $z^* \in [0.8, 0.9]$ for $q_X \in \left[\frac{1}{4}, \frac{3}{4}\right]$. Moreover as one of the errors vanishes (i.e. $q_X \to 0, 1$), $z^*$ approaches $z = 1$ again. The exponent $z^*$ controls the scaling until $\langle S_L(t) \rangle \sim \mathcal{O}(1)$, such that there is approximately only a single remaining qubit to purify. In the thermodynamic limit, the measurement frustration thus gives rise to an anomalous scaling regime with $z^* \neq 1$ which is supplanted by $z = 1$ scaling only after extensive times.

## VI.  $Y$ AND $Z$ ERRORS

In the case of $Y$ and $Z$ measurements ($q_Y + q_Z = 1$), we observe a new scenario of entanglement transitions: two topological phase transitions that take place at different, nonzero values of the error measurement rate, see Fig. 6. The two transitions correspond again to the breaking of the SPT order and the SB order by the errors. In this case, however, the order of breaking them is reversed compared to the previous scenarios. Here again we find a transient dynamical regime with a different dynamical critical exponent $z^* \neq z$. However, in this case $z^* > z$.

Neither $Y$ nor $Z$ measurements alone immediately break the SPT order, and we observe that the SPT order survives throughout the entire stabilizer phase for arbitrary $q_Y + q_Z = 1$. Unlike the $q_Y = 1$ and $q_Z = 1$ limits, however, the SPT order and bulk-symmetry are broken at two separate transitions. Here, the bulk symmetry breaking (SB) transition takes place at a nonzero error measurement rate $1 - p_M^{\text{Stab}}$, which is *smaller* than the critical rate for the SPT order breaking (see Fig. 6). Since the global qubit protected by the XZZX code is encoded via the bulk symmetry, the SB transition is accompanied by a vanishing of the mutual information. In contrast to all previously inspected cases, the SB transition and the vanishing of the mutual information notably are *not* accompanied by any signature of critical behavior in $S_{\text{topo}}^c$. Interpreting a vanishing mutual information as the point where a globally encoded logical qubit is irreversibly destroyed by the errors, this implies that the information loss in the qubit is not signalled by a critical point if $Y$ and $Z$ errors are present simultaneously.

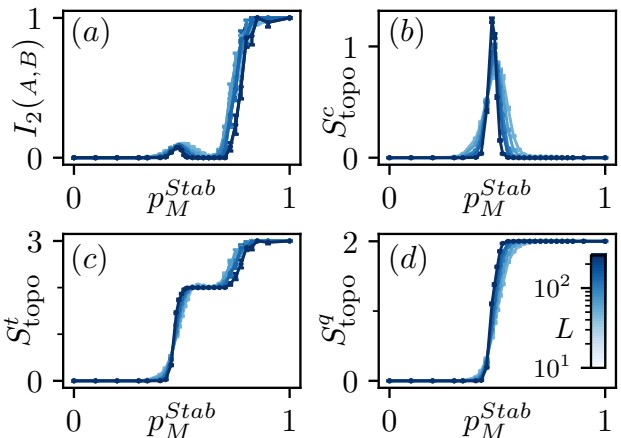

Figure 6. Entanglement measures for system sizes $L = \{48, 64, 96, 128, 256\}$ with $q_Y = q_Z = \frac{1}{2}$. (a) The mutual information between well-separated regions of size $L/8$ vanishes at $p_M^{\text{Stab}} \approx \frac{3}{4}$ where the bulk symmetry is broken. (b) From $S_{\text{topo}}^c$, we observe a critical point at $p_M^{\text{Stab}} \approx 0.466$ with log-law entanglement scaling characterized by finite $\tilde{c}$. (c,d) $S_{\text{topo}}^t$ and $S_{\text{topo}}^q$ show a bulk-symmetry breaking transition which occurs deeper into the stabilizer phase than the SPT breaking transition. It is the SPT breaking transition for which we find log-law entanglement scaling.

The subsequent SPT transition remains near the original phase boundary $p_M^{\text{Stab}} = \frac{1}{2}$ and is accompanied

by a nonzero log-law entanglement scaling, indicating a critical point, consistent with $\nu = \frac{4}{3}$. Interestingly, the magnitude of $\tilde{c}$ along the SPT critical line in the $p_M^Y$—$p_M^Z$ plane is *enhanced* relative to the $q_Y = 1$ and $q_Z = 1$ endpoints. For $p_M^Y = p_M^Z$ at criticality we find $\tilde{c} \approx 4.1 \approx 7.1 \frac{3\sqrt{3}\log(2)}{2\pi}$, as can be seen from the peak value of $S_{\text{topo}}^c$ in Fig. 6. Unlike with $X$ and $Z$ errors, here $\tilde{c}$ varies continuously with $q_Y, q_Z$ without any discontinuous jumps. In this case, $\tilde{c}$ might no longer serve as a universal indicator of the underlying percolation transition. Instead, it might be thought of as counting the (average) number of overlapping cluster states. The full phase diagram in the $p_M^Y$—$p_M^Z$ plane can be found in App. D.

We want to stress that the type of topological phase transitions observed in the previous regimes, i.e., for $q_\alpha = 1$ and for $q_X + q_Z = 1$, have a counterpart in a purely Hamiltonian system. In particular, the measurement-only evolution can be connected to imaginary-time evolution under a corresponding Hermitian Hamiltonian constructed from the stabilizers. The ground state of a $XZZX$-stabilizer Hamiltonian with additional magnetic fields in the $\alpha$-direction[51,52,58] undergoes a topological phase transition with equivalent signatures in the topological entanglement entropies. Due to the inherently different generators of the dynamics, the ground state phase transition corresponds to the one-dimensional Ising universality class instead of 2D bond percolation. However, the breaking of the topological order and the position of the critical point indicate the same topological order in the Hamiltonian and the measurement-only dynamics. However, we emphasize that a separation of the SPT and SB transitions appears to be *unique* to the measurement-only circuit, and to have *no* counterpart in the Hamiltonian setting. The ground state of a $XZZX$-stabilizer Hamiltonian with both $Y$ and $Z$ fields, obtained from exact diagonalization, undergoes only a single transition at which all orders are broken simultaneously (see App. D 1).

In the dynamics at the critical point of the SPT breaking transition, we observe a similar scenario as with $X$ and $Z$ errors. The entanglement growth when starting from a pure state, the entanglement fluctuations in the steady state and the asymptotic purification dynamics all confirm $z = 1$ in the steady state. Again, the transient scaling regime yields a dynamical critical exponent. Unlike the $q_Y = 0$ case, however, the exponent is enhanced $z^* > 1$ compared to the steady state. Around $q_Y = q_Z = \frac{1}{2}$ it reaches a plateau with $z^* \approx 1.17$ (see Fig. 7), and as $q_Y \to 0, 1$ we have $z^* \to 1$.

## VII. CONCLUSION

Here we studied the entanglement dynamics which arise in the one-dimensional XZZX code subject to single site Pauli measurement errors. When only a single type of error is permitted, the dynamics corresponds to the

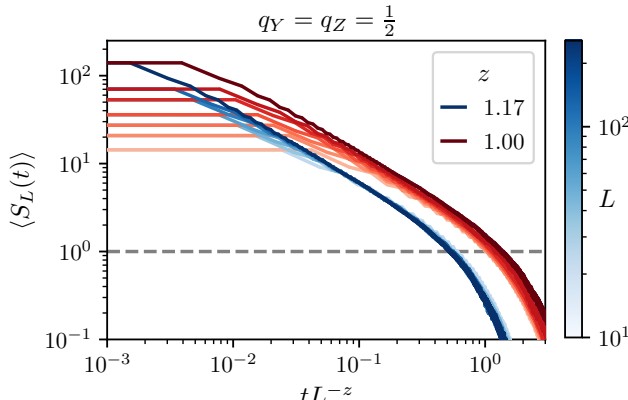

Figure 7. Decay of the residual entropy $\langle S_L \rangle$ for $q_Y = q_Z = \frac{1}{2}$ at the critical point $p_c = 0.46$ for system sizes $L = \{24, 36, 48, 64, 96, 128, 256\}$. Time is rescaled with exponent $z^* = 1.17$ (blue) and $z = 1$ (red). We see a much better data-collapse for $z^* \neq 1$. As with $X$ and $Z$ errors, the transient scaling regime persists until $\mathcal{O}(1)$ qubits remain unpurified (dashed gray lines).

repetition code, admitting an explanation of the entanglement transition in terms of quasi-GHZ clusters and 2D bond percolation. For multiple incommensurate error measurements, the critical behavior still appears to be controlled by 2D bond percolation, although the interplay of measurement frustration and symmetry not only may yield a continuously varying $\tilde{c}$ along the critical line but shifts the relative positions of the SPT and bulk-symmetry breaking transitions. In addition, we observe a separation of the topological phase transitions for $Y$ and $Z$ errors in Sec. VI, which does not posses a ground state counterpart in a translationally invariant Hamiltonian setting. It is an interesting challenge for future work to figure out whether this transition may be unique to the measurement setting or may have a counterpart in an appropriate Hamiltonian, e.g., in the strong-randomness limit

Along the critical line, we find that the pure-state dynamics and asymptotic purification dynamics are characterized by a dynamical critical exponent $z = 1$. However, incommensurate errors yield an extensive scaling regime with anomalous exponent $z^* \neq z$ in the purification dynamics. Very recently, in the 2D toric code subject to only Y errors, $z > 1$ was found[42] at a tricritical point of entanglement transitions. Our results indicate that transient dynamical scaling with a modified critical dynamical exponent may be a general feature of quantum error correcting codes, which correct incommensurate errors. The scaling regime is absent when starting from a pure state, which may hint towards a generally modified dynamical critical behavior of mixed states. As such, it might prevail when the measurements are balanced by a nonzero dephasing rate that drives the system toward a mixed-state close to equilibrium at late times. On the

other hand, when comparing the purification dynamics of an initial mixed state in the measurement setting with the relaxation of an excited state back to equilibrium, the emergence of a transient dynamical critical exponent is reminiscent of the dynamical behavior of prethermal states at a critical point in Hamiltonian or Lindblad dynamics.[62,63] The decrease (increase) of $z^*$ compared to $z$ is consistent with the shift of the critical point to larger (smaller) values of $p_M^{\text{Stab}}$ for $q_Y = 0$ ($q_X = 0$), indicating that both may be traced to a change in the GHZ cluster formation in the presence of incommensurate errors.

## ACKNOWLEDGMENTS

We thank M. Müller, T. Botzung, and S. Diehl for fruitful discussions. KK was supported by the U.S. Department of Energy, Office of Science, National Quantum Information Science Research Centers, Quantum Science Center. MB acknowledges support from the Deutsche Forschungsgemeinschaft (DFG, German Research Foundation) under Germany's Excellence Strategy Cluster of Excellence Matter and Light for Quantum Computing (ML4Q) EXC 2004/1 390534769, and by the DFG Collaborative Research Center (CRC) 183 Project No. 277101999 -project B02.

## Appendix A: Finite-Size Scaling

The critical exponent $p_c$ and exponent $\nu$ are found by finite-size scaling of the various entanglement measures (e.g. $S_{\text{topo}}^{q,c,t}$) as in Fig. 3. Letting $x = (p - p_c)L^{1/\nu}$ and $y$ an entanglement measure (e.g. $I_2$ or $S_{\text{topo}}^{t,q,c}$), under optimal data-collapse, the data $(x_i, y_i)$ fall along a single smooth curve. For sufficiently dense data, each data point $y_i$ should then be well approximated by a linear interpolation of its adjacent data points,

$$\bar{y} = y_{i-1} + \frac{x_i - x_{i-1}}{x_{i+1} - x_{i-1}}(y_{i+1} - y_{i-1})$$
$$= \frac{(x_{i+1} - x_i)y_{i-1} - (x_{i-1} - x_i)y_{i+1}}{x_{i+1} - x_{i-1}}. \quad \text{(A1)}$$

The deviations $(y_i - \bar{y})^2$ then provide a metric for how far from an ideal data collapse is achieved under the rescaling. More precisely, the deviations are normalized against the uncertainty in the data. If $s_i$ is the uncertainty in $y_i$, then we may define the expected variance

$$|\Delta(y - y_i)|^2 = s_i^2 + \left(\frac{x_{i+1} - x_i}{x_{i+1} - x_{i-1}}\right)^2 s_{i-1}^2$$
$$+ \left(\frac{x_{i-1} - x_i}{x_{i+1} - x_{i-1}}\right)^2 s_{i+1}^2 \quad \text{(A2)}$$

and a normalized deviation

$$w_i = \left(\frac{y_i - \bar{y}}{\Delta(y_i - y)}\right)^2. \quad \text{(A3)}$$

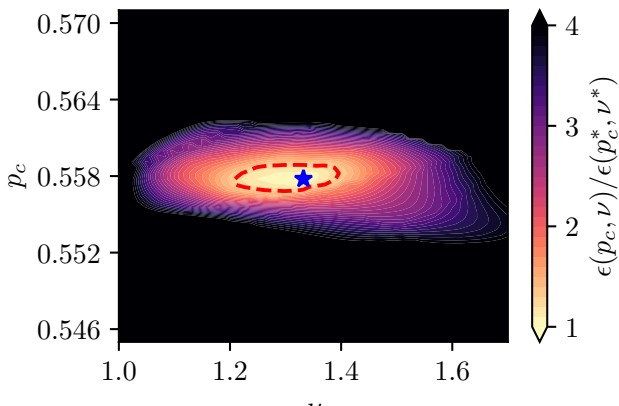

Figure 8. Normalized cost function $\epsilon(p_c, \nu)$ in the $p_c$–$\nu$ plane evaluated for $S_{\text{topo}}^q$ at $q_X = q_Z = \frac{1}{2}$.. The cost is minimized for $(p_c^*, \nu^*) = (0.558, 1.33)$, marked with a blue star. The dashed red contour denotes the region where $\epsilon(p_c, \nu) \leq 1.3\,\epsilon(p_c^*, \nu^*)$, giving an uncertainty estimate of $\pm 0.001$ for $p_c$ and $\pm 0.09$ for $\nu$.

The total deviation

$$\epsilon = \frac{1}{n-2} \sum_{i=2}^{n-1} w_i \quad \text{(A4)}$$

then provides a global cost function which we may minimize with respect to $p_c$ and $\nu$ to find the optimal scaling collapse.[6,18,64] We estimate the critical values $p_c^*$ and $\nu^*$ by minimizing $\epsilon$. Similarly the error in these values is estimated by identifying the region in parameter space where $\epsilon(p_c, \nu) \leq 1.3\,\epsilon(p_c^*, \nu^*)$. Figure 8 gives an example of minimizing the cost function $\epsilon$.

For purification, the functional form of $\langle S_L(t) \rangle$ is known, and so we may find the dynamical exponent $z$ (or $z^*$ in the transient scaling regime) by regression. Fixing a time interval (e.g. $t/L < 1$) we minimize the error with respect to $z$ for fitting a power-law to the residual entropy density. Similarly at late times one can minimize the error with respect to fitting an exponential decay. As with the cost function $\epsilon$, the error in the estimate of $z^*$ can be approximated by a threshold (e.g. $1.3$ times the minimum fitting error). This gives comparable results to the linear interpolation cost function approach used for finding $(p_c, \nu)$. In Fig. 9 we show the normalized fitting error for the purification dynamics in the transient (power-law) regime and the late-time (exponential) regime, showing a distinct $z^* \neq 1$ which gives way to $z \approx 1$ at asymptotically late times.

## Appendix B: Additional Data for Unique Pauli Errors

Here we provide supporting figures for the case of unique Pauli errors. In Fig. 10 we show explicitly the

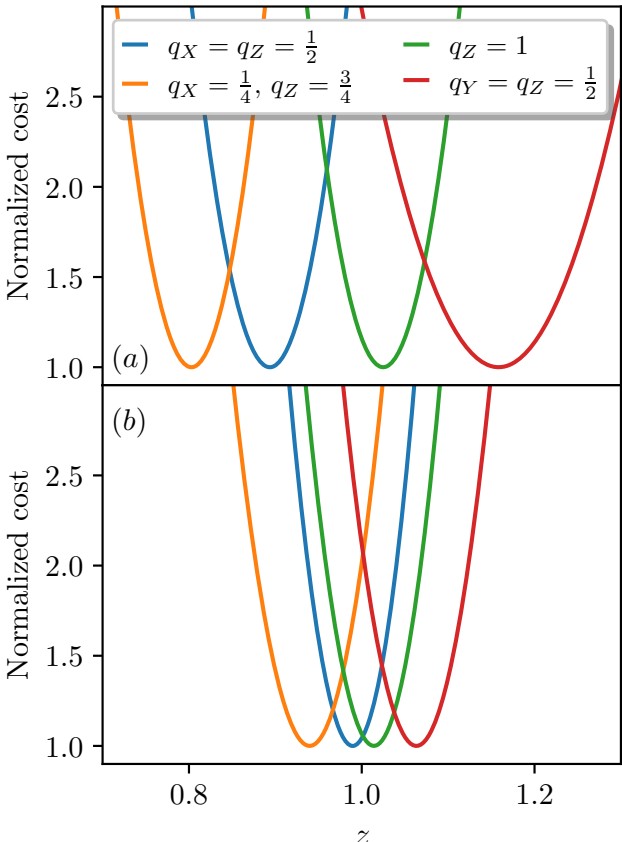

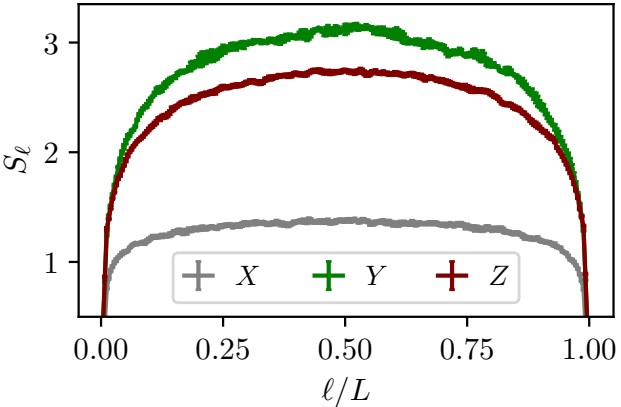

Figure 10. Entanglement entropy of subregion $A = [1, \dots, \ell]$ as a function of $\ell$ for unique errors ($q_\alpha = 1$) at criticality ($p_M^{Stab} = \frac{1}{2}$) with system size $L = 256$.

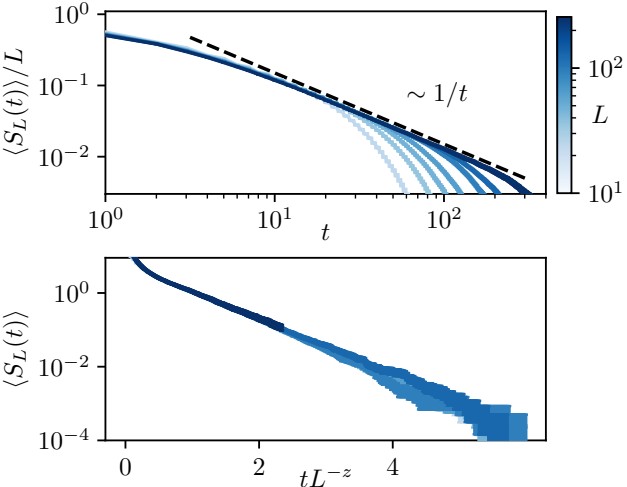

Figure 11. Purification dynamics for $Y$ errors only with system sizes $L = \{24, 36, 48, 64, 96, 128, 256\}$. (top) The average residual entropy density, at intermediate times decays as $1/t$ (dashed black line), consistent with $z = 1$. (bottom) The residual entropy at late times exhibits data collapse under rescaling $t \to tL^{-z}$ with $z = 1$.

Figure 9. Normalized fitting cost for the dynamical exponent $z$ via purification at criticality. Costs are normalized against the minimum cost, which determines $z^*$. (a) Fitting cost for times where $\langle S_L(t) \rangle > 1$. Here we fit a linear relationship between $\log(tL^{-z})$ and $\log(\langle S_L(t) \rangle)$ to capture the power-law purification. (b) Fitting cost for late times where $\langle S_L(t) \rangle < 1$ Here we fit a linear relationship between $tL^{-z}$ and $\log(\langle S_L(t) \rangle)$ to capture the exponential purification.

logarithmic entanglement growth at the critical point. This confirms the relative values of $\tilde{c}$ indicated by the peak values of $S_{\text{topo}}^c$ in Fig. 1d.

In Fig. 4a, we show that the purification dynamics with only $X$ or only $Z$ errors is consistent with $z = 1$ at all times. In Fig. 11, we verify that this is also the case when only $Y$ errors are allowed.

**Appendix C: Additional Data for $X$ and $Z$ Errors**

Here we provide supporting figures and data for the case of $X$ and $Z$ errors. In Fig. 12 we show the phase diagram in the $p_M^Z$—$p_M^X$ plane measured via $S_{\text{topo}}^t$ and $S_{\text{topo}}^c$. This provides an exact numerical verification of the schematic phase diagram presented in Fig. 1a. Furthermore it provides the values of $\tilde{c}$ along the transition, including the discontinuous jump at $q_Z = 1$. We note that the shifting of the critical point is *not* symmetric

about $q_X = q_Z = \frac{1}{2}$. Rather, the stabilizer phase is reduced in area more appreciably for larger $q_Z$.

Fig. 13 shows the logarithmic growth of the half-chain entanglement entropy at criticality as a function of system size. The dashed line corresponds to $\tilde{c}_t = \tilde{c}$, corroborating the claim that $z = 1$.

For pure states we found a transition with critical exponent $\nu = \frac{4}{3}$. The steady-state of the purification dynamics reproduces the same entanglement transition as is found in the pure-states. This can be seen in Fig. 14, which shows the critical scaling of the steady-state residual entropy $\langle S_L \rangle$ at fixed time $t = 10^3$.

As we note in Sec. V, within a neighborhood of the

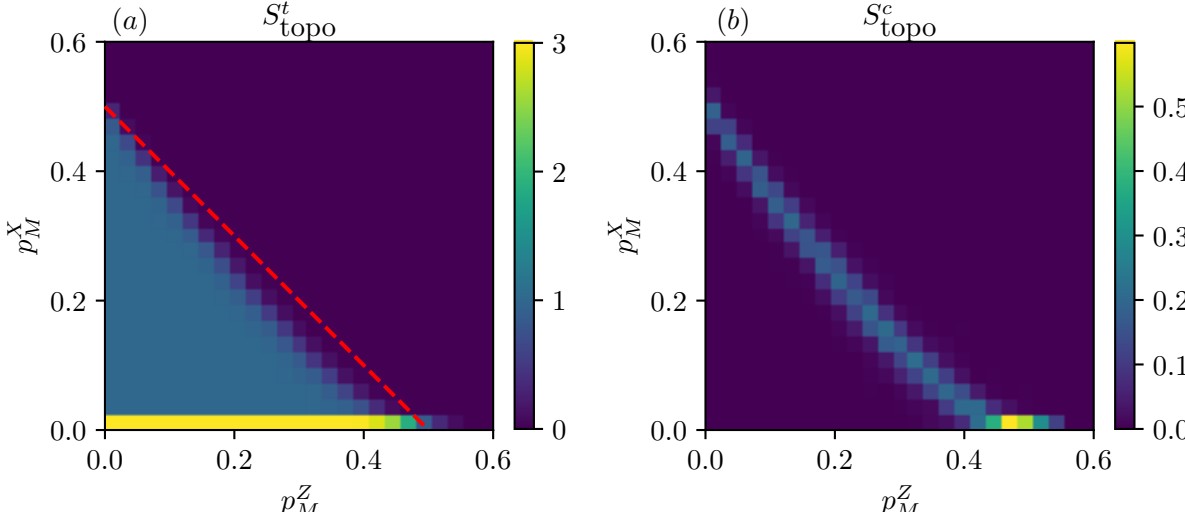

Figure 12. Phase diagram in the $p_M^X$—$p_M^Z$ plane, with $p_M^Y = 0$, for $L = 128$. (a) $S_{\text{topo}}^t$ reproduces the schematic phase diagram in Fig. 1a, where SPT order survives only when $p_M^Z = 0$. The dashed red line indicates $p_M^{Stab} = \frac{1}{2}$, showing that with $X$ and $Z$ measurements the stabilizer phase is *reduced* in area. (b) $S_{\text{topo}}^c$ reveals both the phase boundary and the log-law coefficient $\tilde{c}$. We clearly observe a discontinuous jump at $q_Z = 1$, with $\tilde{c}$ remaining very near to the percolation value $\frac{\sqrt{3}\log(2)}{2\pi}$ everywhere else along the transition.

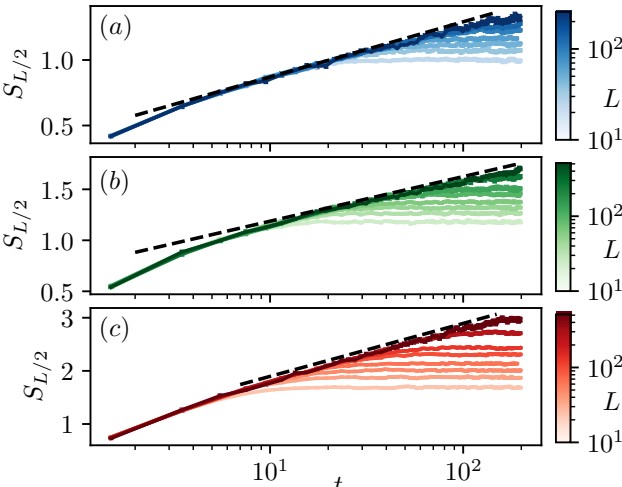

Figure 13. Logarithmic growth of the half-chain entanglement entropy $S_{L/2}(t)$ at criticality with (a) $q_X = 1$, (b) $q_X = q_Z = \frac{1}{2}$, and (c) $q_Z = 1$. Here we average between odd and even layers of the circuit to remove the finite average difference in entanglement between subsequent steps owing to the alternating layer structure of the circuit. The dashed black line corresponds to $z = 1$. The data represent the average of between $10^3$ and $10^4$ individual trajectories for system sizes $L = \{24, 36, 48, 64, 96, 128, 256, 512\}$.

critical point $(p - p_c)L^{1/\nu} \ll 1$, the exponent $z^*$ describing the transient purification regime is robust against small perturbations in $p_M^{\text{Stab}}$. In Fig. 15 we show this for $p_M^{Stab} \in [0.53, 0.57]$ with $z^* = 0.85$ giving a good data-

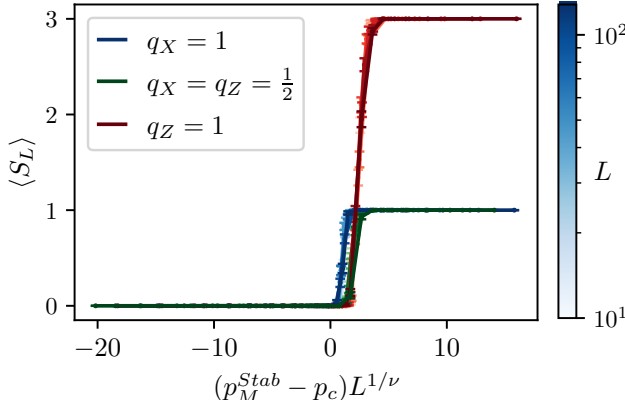

Figure 14. Average residual entropy $\langle S_L(t) \rangle$ at time $t = 10^3$ for system sizes $L = \{16, 24, 36, 48, 64, 96, 128\}$. The data collapse under a rescaling of the measurement strength with critical exponent $\nu = \frac{4}{3}$, as in the pure-state transition.

collapse up until there are $\mathcal{O}(1)$ remaining qubits to be purified.

## Appendix D: Additional Data for $Y$ and $Z$ Errors

This section consists of supplementary data and figures for the case where both $Y$ and $Z$ errors are allowed. In Fig. 18, we show the phase diagram via $S_{\text{topo}}^t$ and $S_{\text{topo}}^c$ in the $p_M^Z$—$p_M^Y$ plane as measured via $S_{\text{topo}}^t$, providing a numerical verification of the schematic version shown in

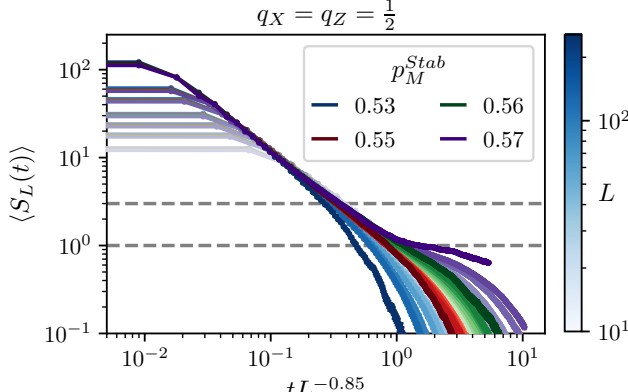

Figure 15. Decay of the residual entropy $\langle S_L \rangle$ in the transient regime near criticality for $L = \{24, 36, 48, 64, 96, 128, 256\}$. Here time is rescaled with exponent $z^* = 0.85$. The exponent $z^*$ is insensitive to $p_M^{\text{Stab}}$ within a narrow range of the critical point $p_c$ up until times where $\langle S_L(t) \rangle$ is $\mathcal{O}(1)$ (dashed grey lines at 1 and 3).

Fig. 1b. We see here explicitly that the stabilizer phase is enlarged relative to the case with only one type of allowed error. From $S_{\text{topo}}^c$ we also observe the enhancement of the prefactor of the logarithmic entanglement growth in the presence of measurement frustration.

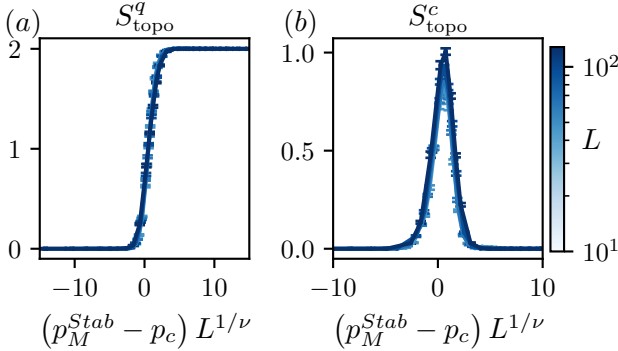

Figure 16. Scaling collapse for $S_{\text{topo}}^q$ and $S_{\text{topo}}^c$ at the SPT-breaking transition with $q_Y = q_Z = \frac{1}{2}$. Data are consistent with an exponent $\nu \approx \frac{4}{3}$ and critical point $p_c \approx 0.47$.

As we discuss in Sec. VI, we find logarithmic entanglement scaling only at the SPT-breaking transition, whereas for $X$ and $Z$ errors this was found at the bulk-symmetry breaking transition. Nonetheless, Fig. 16 shows the scaling collapse near the critical point for $q_Y = q_Z = \frac{1}{2}$ still gives the percolation exponent $\nu = \frac{4}{3}$. We note also that the vanishing of the mutual information at the bulk-symmetry breaking transition appears consistent with an exponent $\nu = \frac{4}{3}$ despite not being

accompanied by a log-law entanglement scaling.

### 1. Exact Diagonalization

Here we comment comment briefly on the origin of the separation of the SPT and bulk-symmetry breaking transitions. For $X$ and $Z$ errors, the SPT order is naturally broken by any finite $p_M^X$. On the other hand, for $Y$ and $Z$ errors we have shown that the bulk-symmetry and SPT order break at two distinct but finite error measurement probabilities. We are interested in identifying whether the splitting of the SPT and bulk-symmetry transitions is a feature unique to the measurement scenario. Let us consider the uniform $XZZX$ model with external field,

$$ H = J \sum_i X_i Z_{i+1} Z_{i+2} X_{i+3} + \sum_i (h_Y Y_i + h_Z Z_i) . \quad \text{(D1)} $$

Fixing $J = 1$ and $h_Z = \frac{1}{4}$, we vary $h_Y$ and extract the entanglement measures in the ground state via exact diagonalization. As seen in Fig. 17b, the bulk symmetry and the SPT order both break at the same value of $h_Y$. Moreover, the finite-size scaling analysis in Fig. 17a gives an estimated critical point $h_Y \approx 1.0 \pm 0.1$ and critical exponent $\nu \approx 0.9 \pm 0.4$ consistent with an Ising-type transition. This then suggests that the separation of the two transitions may arise either (i) purely from the randomness in measurements, or (ii) genuinely from the measurement. For the former, we might consider quenched disorder in the infinite-randomness limit. We leave further disambiguation of the origin of this separation of transitions to a future work.

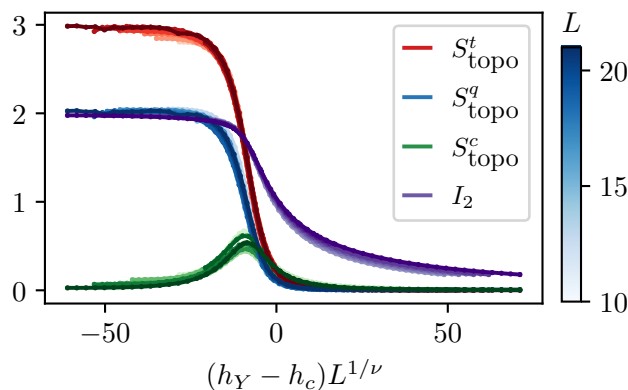

Figure 17. Data collapse for entanglement measures calculated in the ground state of the uniform $XZZX$ Hamiltonian (Eq. (D1)) with open boundary conditions, $J = 1$, and $h_Z = \frac{1}{4}$ for system sizes $L \in [12, 21]$. The entanglement measures show good data collapse under rescaling of $h_Y$ to $(h_Y - h_c)L^{1\nu}$ with $h_c \approx 0.94$ and $\nu \approx 0.73$. Moreover, all data suggest that the SPT and SB transitions occur simultaneously in this model.

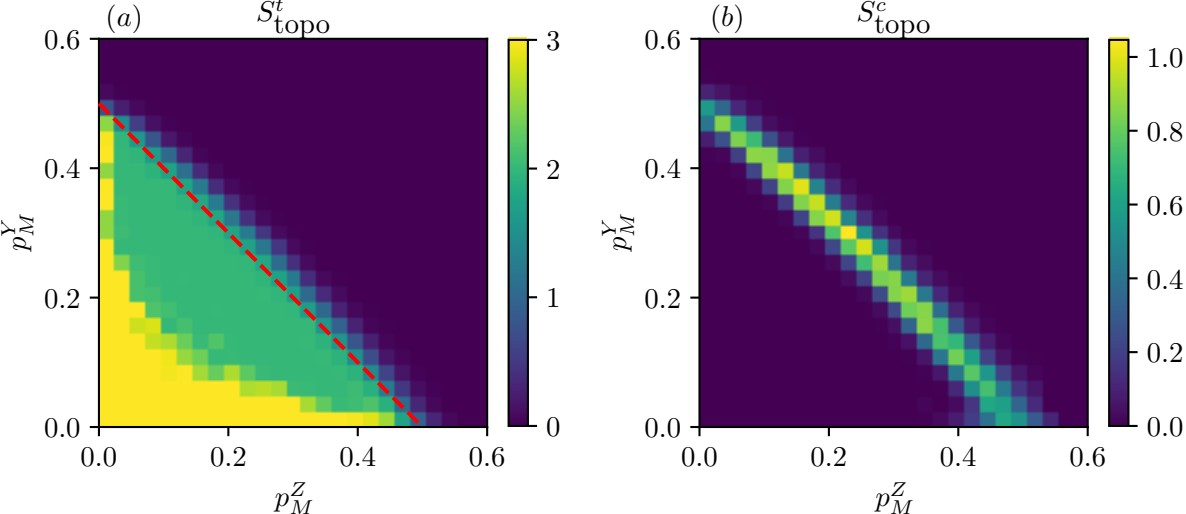

Figure 18. Phase diagram in the $p_M^Y$—$p_M^Z$ plane, with $p_M^X = 0$, for $L = 128$. (a) $S_{\text{topo}}^t$ reproduces the schematic phase diagram in Fig. 1b, with SPT order extending through the entire stabilizer phase while the bulk-symmetry breaks at a smaller but finite error measurement probability. The dashed red line indicates $p_M^{Stab} = \frac{1}{2}$, showing that with $Y$ and $Z$ measurements the stabilizer phase is *enlarged* in area. (b) $S_{\text{topo}}^c$ reveals both the phase boundary and the log-law coefficient $\tilde{c}$. Here we find no discontinuous jumps. Furthermore we clearly observe an enhancement of $\tilde{c}$ for mixed errors. Due to finite size $L$ and grid-spacing, the values here underestimate slightly those reported with a finer mesh, as in the main text (e.g. Fig. 6).

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
