# Peer review of "Topological order and entanglement dynamics in the measurement-only XZZX quantum code"

_SciPost Physics_

## Round 2 · Referee Report · Anonymous · 2022-5-4

Strengths

The manuscript:
1. is well-written;
2. provides results on timely topic.

Furthermore, the authors provide a code which simplify the reproducibility of the numerical data.

Weaknesses

1. Lack of analysis for finite size scaling and data collapses;
2. The main observations are mostly observational and with a limited discussion;
3. Limited novelty.

Report

The paper "Topological order and entanglement dynamics in the measurement-only XZZX quantum code" by K. Klocke and M. Buchhold discusses an implementation of measurement-only dynamics, where a generalization of the (error-correcting) [[5,1,3]] code (the XZZX-model) is interspersed with local projections of $X, Y, Z$ operators (hereby, as in the manuscript, denoted errors).

Alone, the code presents symmetry-protected topological (SPT) and symmetry-breaking phases (SB). The presence of errors affects the phase diagram, which the authors explore in terms of different control parameters:
1. the probability of projecting onto the code $p_\mathrm{stab}$, which act on the system on odd (discrete) times;
2. the probability of errors $q_\alpha$ ($\alpha=X,Y,Z$).
For some $q_\alpha=1$ (hence when only a type of error is present), the considered framework reduces, up to minor modifications, to the ones discussed in Ref.[41] and Ref.[44]. Here the transition lies in the 2D percolation critical point universality class, with the correlation length critical exponent $\nu=4/3$ and the dynamical critical exponent $z=1$.

The novelty of the work is discussed in Sec. 5 and Sec. 6, where the authors discuss, respectively, the simplex $q_x+q_z=1$ and $q_y+q_z=1$. The following are the main contributions of the paper:

1. The combined presence of non-communing errors give rise in both scenarios to frustration effects, resulting in a dynamical transient phase with $z^*\neq 1$ when the initial state is fully mixed, and as captured by the purification dynamics. This transient regime is claimed to hold for periods $T\sim O(L)$ ($L$ the system size), which hence extends for an extensively large time; at later time the expected $z=1$ is recovered.

2. The case of some $q_\alpha=1$ and $q_x+q_z=1$ have an analogy to the ground state phase transitions of analog $XZZX$ Hamiltonians with additional external field along the $\alpha$ directions, whereas the $Y$ and $Z$ case (Sec. 6) does not have such an analogy. Indeed, the phase diagram of the circuit in this limit has two separate transitions (one of SB type and the other of SPT type), while the Hamiltonian associated to the model (Appendix C, Eq.(3)) does not: the authors claim only a combined SB $\textit{and}$ SPT transition occurs in this case.

Overall the paper is well written, and discuss a new instance (model) for a measurement-only phase transition. In this present form, I would recommend the paper to be published in SciPost Physics Core.

However, the present form of the manuscript does not meet the stricter conditions of SciPost Physics, for which the authors applied for. Below I will give the list of criticisms I find the authors should resolve in a possible future resubmission.

1. $\textit{Lack of finite size scaling analysis and data collapse}$. I don't understand how the analysis for the various exponents, in particular of $z$ are done in the paper. From Fig. 5 and Fig. 7 I see the point at which the curves bend (hence the collapse for $z=z^*$ fails) shifts toward the left for larger system sizes (hence if occurs at earlier times). In particular, it may be that the transient extends for times $T\sim O(L)$ as a result of finite size effects, whereas in reality $\lim_{L\to \infty}T=T^\star<\infty$. Without a systematic analysis of this point, I'm not convinced the "pre-thermal" phase extends for extensive times.

Furthermore, it would be desirable to include a second plot which compares Fig. 5 and Fig. 7 with the case $z=1$ (for instance, adding a second subplot with $z=1$ and the same lines). In these plots I would mostly discuss the case of the critical point, which is the relevant one to understand the dynamical critical exponent: this would avoid having many lines which give less clear figures).
(My doubts come also from the fact that $z^*\simeq 0.85\div 0.9$ and $z^\star\simeq 1.25$ are close to $z=1$ within possible fitting/optimization errors.)

2. $\textit{Analogy with ground state Hamiltonian}$. For the $q_y+q_z=1$ case, the authors find evidence of two separate transitions. They discuss that this phenomenology (proper of the circuit model) does not have a simple analog in ground state Hamiltonian models. They consider Eq.(3) and suggest that only a combined SB and SPT transition occurs. However from Fig.15 I only see a system size considered $L=18$. How can the authors discriminate if one or two transitions may occur without a finite size scaling?

On a more general ground, the link between the XZZX Hamiltonian with external field and the present dynamical scenario seems not totally well justified (and more of a coincidence). The measurements should introduce a non-unitarity element, which may be more suitably encoded in a non-hermitian Hamiltonian framework (e.g. imaginary field $\alpha$ transverse fields, see for instance, the PRL of Gopalakrishnan and Gullans, or the statistical mappings of Altman's group). Without a better motivation, I don't see the relevance of this comment/ analysis.

3. $\textit{Limited relevance/novelty.}$ Lastly, I think that the paper, in the present form, provides slight and limited advances to the field of measurement-induced criticality. The paper is (almost completely) observational, with methods and phenomenology previously discussed in a variety of works (e.g. Ref.[16,41-45]). For instance there is a very similar model discussed in the seminal Nature of A. Lavasani, et al (the XZX model): to me the extension to the XZZX model and considering various single site measurements seems only "filling the details", compared with the previous literature.

Requested changes

$\textbf{For a resubmission in SciPost Physics:}$
1. Include an extensive finite size scaling and data collapse analysis for the various exponent obtained. In particular, how robust is the finite size scaling of the transient regime $z^*$? Does this changes/shifts for larger system sizes to earlier times? Update Fig 5 and Fig 7 with a comparison with $z=1$ and considering preferably $\textbf{only}$ the critical point.

2. Motivate the analogy with ground state Hamiltonians and include the finite size scaling for fig 15.

3. Clarify/justify in what/why this paper introduces substantial novelty to the field of measurement-induced criticality, as compared to the previous works.

$\textbf{For SciPost Physics Core:}$
I would suggest the paper for publication even in the present form. However I would encourage the authors to include a discussion on the finite size scaling and the data collapse presented.

$\textbf{Auxiliary improvement:}$
The authors may consider adding a discussion on the Gottesman-Knill and the simulation methods used in the paper. Although this is partially available in the comments of the code, adding such an appendix would make the paper more self-contained.
Additionally, it would be better to upload the code to some archive (which is more resilient that a personal github page, and has also a DOI in case of future references).

---

## Editorial Decision

editor-in-charge_assigned